

# Herbarium specimens can reveal impacts of climate change on plant phenology; a review of methods and applications

Casey A. Jones and Curtis C. Daehler

Department of Botany, University of Hawaii at Manoa, Honolulu, HI, United States of America

## ABSTRACT

Studies in plant phenology have provided some of the best evidence for large-scale responses to recent climate change. Over the last decade, more than thirty studies have used herbarium specimens to analyze changes in flowering phenology over time, although studies from tropical environments are thus far generally lacking. In this review, we summarize the approaches and applications used to date. Reproductive plant phenology has primarily been analyzed using two summary statistics, the mean flowering day of year and first-flowering day of year, but mean flowering day has proven to be a more robust statistic. Two types of regression models have been applied to test for associations between flowering, temperature and time: flowering day regressed on year and flowering day regressed on temperature. Most studies analyzed the effect of temperature by averaging temperatures from three months prior to the date of flowering. On average, published studies have used 55 herbarium specimens per species to characterize changes in phenology over time, but in many cases fewer specimens were used. Geospatial grid data are increasingly being used for determining average temperatures at herbarium specimen collection locations, allowing testing for finer scale correspondence between phenology and climate. Multiple studies have shown that inferences from herbarium specimen data are comparable to findings from systematically collected field observations. Understanding phenological responses to climate change is a crucial step towards recognizing implications for higher trophic levels and large-scale ecosystem processes. As herbaria are increasingly being digitized worldwide, more data are becoming available for future studies. As temperatures continue to rise globally, herbarium specimens are expected to become an increasingly important resource for analyzing plant responses to climate change.

Corresponding author
Casey A. Jones, jonesc22@hawaii.edu

## INTRODUCTION

Carl Linnaeus pioneered the study of phenology when he outlined methods for investigating associations between flowering and climate in the 1700s (*Linnaeus, 1751*; *Von Linné, 2003*; *Puppi, 2007*). Around 1850, Charles Morren introduced the term "phenology" to describe his observational studies of yearly flowering (*Morren, 1853*; *Demarée & Rutishauser, 2009*). Early field studies of plant phenology have been thoroughly reviewed by *Van Schaik, Terborgh & Wright (1993)*, *Fenner (1998)* and *Forrest & Miller-rushing (2010)*. Long-term observations in field studies have provided a valuable resource for analyzing phenological

responses to recent climate change (*Walther et al., 2002*; *Parmesan & Yohe, 2003*). A growing need for historical data that allows for the exploration of ecological implications of climate change prompted researchers to look to herbarium specimens. A few phenology studies such as *Borchert (1996)* and *Rivera & Borchert (2001)* used herbarium specimens to study flowering periodicity, but not in the context of climate change. The first study to use herbarium specimens to understand phenological responses to climate change was published in 2004 by *Primack et al. (2004)*. *Primack et al. (2004)* analyzed 372 specimen records (1885–2002) and found flowering had advanced approximately eight days over the last century. Between 2004 and 2017, more than 30 studies were published using herbarium specimens to examine changes in phenology in response to climate change.

The most common approach found in studies using herbarium specimens follows the procedure set by *Primack et al. (2004)*. This can be summarized as collecting Julian dates from herbarium specimens, collecting long-term temperature data from an independent source, and then using regression analyses to analyze correlations between Julian dates, temperatures and time (*Primack et al., 2004*; *Miller-Rushing et al., 2006*; *Gallagher, Hughes & Leishman, 2009*; *Robbirt et al., 2011*; *Gaira, Dhar & Belwal, 2011*; *Molnár et al., 2012*; *Panchen et al., 2012*; *Park, 2012*; *Primack & Miller-rushing, 2012*; *Li et al., 2013*; *Calinger, Queenborough & Curtis, 2013*; *Hart et al., 2014*; *Rawal et al., 2015*; *Park & Schwartz, 2015*). *Primack et al. (2004)* recorded the date of collection from each herbarium specimen and then extracted Julian dates from the collection dates. A Julian date is a value between 1 and 365 corresponding to the day of year when the specimen was collected. Linear regression models are also the most widely used statistical models in field studies investigating flowering phenology (*Zhao et al., 2013*).

An early criticism of using herbarium specimens was that plant parts preserved as herbarium specimens might not have been collected during their peak flowering season, potentially biasing interpretations (*Lamoureux, 1973*). *Daru, Van der Bank & Davies (2017)* also found spatial, temporal, trait, phylogenetic, and collector biases among herbarium specimen samples. *Daru, Van der Bank & Davies (2017)* concluded that while some of these biases can be accounted for using statistical approaches, future herbarium collections should focus on filling large gaps in the data. Other studies have found that large sample sizes afforded by herbarium specimens, and the use of mean flowering times (mean of Julian dates), could yield valid inferences, even if specimens were not collected at the time of peak flowering (*Primack et al., 2004*; *Bertin, 2015*). Collector bias and plant size choice have also been overcome by statistical analyses when mean flowering times were used as the variable of interest, rather than the date of first-flowering (*Robbirt et al., 2011*; *Davis et al., 2015*).

Most of the studies we reviewed used two types of linear regression models to show evidence of associations between phenology and climate change (Table 1). These studies regressed flowering day on temperature (82%) and flowering day on year (64%) (Table 1). These studies have primarily been conducted with specimens from herbaria in temperate latitudes such as the Eastern Himalayas (*Gaira, Dhar & Belwal, 2011*; *Li et al., 2013*; *Gaira et al., 2014*; *Hart et al., 2014*), Southern Australia (*Gallagher, Hughes & Leishman, 2009*; *Rawal et al., 2015*), Northern Europe (*Robbirt et al., 2011*; *Diskin et al., 2012*; *Molnár et al.,*

**Table 1 Methods of studies.** The column "Flw Day ~ Temp" represents studies that conducted a type of regression analysis with flowering day (Flw Day) as the dependent variable and temperature average (temp) or year as the independent variable. The "$\Delta \bar{x}$" symbol represents studies that analyzed a difference in the mean flowering day between historic and current time period groups rather than using a type of regression analysis.

| Species | Specimens | Specimen per species | Authors | Year | Geographic region | (flw ~ temp) | (flw ~ year) |
|---|---|---|---|---|---|---|---|
| 1 | 117 | 117 | Gaira et al. | 2011 | Eastern Himalayas | | x |
| 1 | N/A | N/A | Gaira et al. | 2014 | Eastern Himalayas | x | x |
| 1 | 192 | 192 | Robbirt et al. | 2011 | Northern Europe | x | |
| 5 | 158 | 32 | Rawal et al. | 2015 | Southern Australia | x | x |
| 5 | 540 | 108 | Diskin et al. | 2012 | Northern Europe | x | x |
| 20 | 371 | 19 | Gallagher et al. | 2009 | Southern Australia | x | x |
| 20 | 1,108 | 55 | Davis et al. | 2015 | North America | x | x |
| 28 | 1,587 | 57 | Panchen et al. | 2012 | North America | x | x |
| 36 | 460 | 13 | Hart et al. | 2014 | Eastern Himalayas | x | |
| >37 | 372 | 10 | Primack et al. | 2004 | North America | x | x |
| 39 | 216 | 6 | Lavoie & Lachange | 2006 | North America | | x |
| 39 | 5,424 | 139 | Molnár et al. | 2012 | Northern Europe | | x |
| 41 | 909 | 22 | Li et al. | 2013 | Eastern Himalayas | x | x |
| 42 | 142 | 3 | Miller-Rushing et al. | 2006 | North America | x | x |
| 43 | N/A | N/A | Primack & Miller-Rushing | 2012 | North America | | x |
| 87 | N/A | N/A | Neil et al. | 2010 | North America | | x |
| 141 | 5,053 | 36 | Calinger et al. | 2013 | North America | x | |
| 186 | 30,000 | 161 | Bertin | 2015 | North America | | $\Delta \bar{x}$ |
| 370 | 1,125 | 3 | Searcy | 2012 | North America | | $\Delta \bar{x}$ |
| 1,185 | 5,949 | 5 | Park | 2012 | North America | | x |
| >1,700 | 19,328 | 11 | Park | 2014 | North America | x | |
| 24,105 | 823,033 | 34 | Park & Schwartz | 2015 | North America | x | x |

*2012*), and North America (*Primack et al., 2004*; *Lavoie & Lachance, 2006*; *Miller-Rushing et al., 2006*; *Primack & Miller-Rushing, 2009*; *Neil, Landrum & Wu, 2010*; *Panchen et al., 2012*; *Park, 2012*; *Primack & Miller-rushing, 2012*; *Searcy, 2012*; *Calinger, Queenborough & Curtis, 2013*; *Park, 2014*; *Park & Schwartz, 2015*; *Bertin, 2015*; *Davis et al., 2015*). Although studies by *Borchert (1996)* and *Zalamea et al. (2016)* analyzed flowering periodicity in tropical plants using herbarium specimens, we found no study to date that has used herbarium specimens to analyze effects of recent climate change in a tropical region. In this review, we examined how studies chose sample sizes, flowering specimens, temperature averages and geographical scale in their analyses. We also examined how these studies validated the use of herbarium specimens and we provide suggestions for methods to be used in future studies.

## Survey methodology

Between 2015 and 2017, we compiled and reviewed studies that used herbarium specimens to assess climate change and flowering phenology. We searched Web of Science (1900—present), JSTOR (1665—present) and Google Scholar for studies containing the terms herbarium, specimen, phenology, and climate change. The methods of each study
were reviewed for; sample size, determining flowering status of specimens, approach to determining temperatures, geographic variation, and any validations of the use of herbarium specimens (e.g., comparisons to field observations). Studies and methods were then categorized and a synthesis of each category is discussed; sample sizes and regression methods were also summarized (Table 1).

## Specimen sample sizes

Sample size, or the number of specimens used per species, varied across studies (Table 1). The minimum number of specimens used per species was occasionally as low as two or three records (*Searcy, 2012*). Some studies using herbarium data have set a minimum number of herbarium specimens per species or a minimum time range for collections in order to more accurately estimate phenologies and change over time. *Calinger, Queenborough & Curtis (2013)* and *Gallagher, Hughes & Leishman (2009)* set a minimum of 10 specimens in order to meet statistical assumptions of different models. *Molnár et al. (2012)* eliminated a species from analyses because collections only yielded dates across an eight year time span. *Park & Schwartz (2015)* eliminated species with records that spanned less than three years. *Neil, Landrum & Wu (2010)* organized species into functional groups (spring ephemerals, spring shrubs, fall ephemerals, winter-spring ephemerals, and winter-spring shrubs) in order to overcome the problem of low sample sizes for each species but found that responses of individual species varied greatly within functional groups.

Several studies found sample size had a greater influence on first-flowering estimates than on mean flowering estimates. *Miller-Rushing & Primack (2008)* used field data and found that small sample sizes led to biased estimations of first-flowering dates, but mean flowering day was not biased by sample sizes. *Moussus, Julliard & Jiguet (2010)* investigated sample sizes by simulating 10 known phenological estimators, such as mean flowering day and first-flowering date. After comparing known phenological shifts from simulated sample data with shift estimations from models using the same data, *Moussus, Julliard & Jiguet (2010)* concluded that first-flowering dates were inaccurate because they showed much greater differences in comparisons than mean flowering day. Low sample sizes prompted *Bertin (2015)* to provide a detailed analysis of how sample size affected mean, median, range, early flowering and late flowering summary statistics. In random simulations comparing sample sizes, mean flowering day values deviated less than five days for species with as few as four samples (*Bertin, 2015*). *Bertin (2015)* concluded that the mean was a more robust measure of phenology than other estimators of early flowering. *Bertin (2015)* also showed that by increasing the sample size to 20, mean flowering times deviated only one to two days. A recent study by *Pearse et al. (2017)* used a Weibull distribution to estimate the start of the process of flowering rather than using only first-flowering observations. *Pearse et al. (2017)* showed that by controlling for differences in sampling, first-flowering, peak-flowering (median) and cessation of flowering show similar changes over time in response to climate change. The model used by *Pearse et al. (2017)* was also shown to be consistent with changes in mean-flowering from a separate sample using an early time period.

Larger sample sizes may be required if phenology varies across a species' geographic range. In order to analyze species distributions using herbarium specimens, *Van Proosdij et al. (2016)* found that the minimum number of herbarium specimens sampled should be between 14 and 25 depending on the geographical range of the species. The *Van Proosdij et al. (2016)* study used simulated species to assess the minimum herbarium samples required for acceptable model performance in both virtual and real study areas. Some species with narrow geographical ranges could be modeled with as few as 14 herbarium records while wide ranging species could be satisfactorily modeled with a minimum of 25 records (*Van Proosdij et al., 2016*). Based on these studies, we recommend caution when interpreting results from samples sizes with fewer than 30 records (*Miller-Rushing & Primack, 2008*; *Moussus, Julliard & Jiguet, 2010*; *Bertin, 2015*). The average sample size across studies in this review was about 55 records per species (Table 1). We also recommend using the mean flowering day of year rather than averages of first flowering dates (*Calinger, Queenborough & Curtis, 2013*; *Gallagher, Hughes & Leishman, 2009*; *Pearse et al., 2017*).

## Determining flowering status of specimens

Some studies have simply recorded the presence or absence of flowers from herbarium specimens as an indicator of flowering, but other studies have used more detailed criteria to assess flowering status on specimens. *Haggerty, Hove & Mazer (2012)* provided a primer to assist researchers with collecting data from herbarium specimens. *Haggerty, Hove & Mazer (2012)* suggested researchers assign a phenophase for each specimen, such as pre-flowering, first-flowering or peak flowering. *Haggerty, Hove & Mazer (2012)* also noted that researchers must assume the stem on the herbarium sheet represents the flowering phenophase for the entire plant. Past studies, such as *Diskin et al. (2012)*, have used a scoring system from 1 to 5 to categorize phenophase stages raging from "no flowers" to "end of fruiting" on each specimen. *Diskin et al. (2012)* categorized flowering as 50% of buds open on the specimen. *Calinger, Queenborough & Curtis (2013)* also categorized flowering as 50% of flower buds in anthesis to ensure that the samples were in peak flowering. For a species with an inflorescence, *Davis et al. (2015)* only counted specimens as flowering if greater than 75% of flowers were open. Standardization of phenological terms remains a core challenge of mining phenological data (*Willis et al., 2017*). Initiatives such as the Plant Phenology Ontology (PPO) working group are currently structuring phenological terms for more uniform application across studies (*Willis et al., 2017*).

Studies in temperate regions have used varying methods to determine flowering status for species with long flowering durations. For example, *Molnár et al. (2012)* and *Bertin (2015)* excluded species that flowered outside of the peak flowering season of the region, defined as the period from late-spring to early-summer. *Molnár et al. (2012)* removed one species because its peak flowering date was in September and focused on 40 other taxa that had flowering peaks from in spring and early-summer. The excluded species was a strong outlier and it was suggested that autumn climate events may affect species differently than spring climate events (*Molnár et al., 2012*). *Park (2012)* also removed outlier records when flowering records fell outside the peak regional flowering season. Flowering records before Julian day 45 and after Julian day 310 were removed from analyses to reduce biases caused
by winter flowering species. Additionally, *Park (2012)* removed records that were 150 days after the median flowering date for each species to reduce errors caused by any second flowerings that can happen in autumn months. Several other studies removed taxa with long flowering durations to reduce variance among species. *Bertin (2015)* excluded native weedy species with flowering durations from spring to fall. *Gallagher, Hughes & Leishman (2009)* only used species with a flowering duration of less than three months. *Panchen et al. (2012)* chose to use only species with clear beginning and ending points to investigate long and short flowering duration. *Panchen et al. (2012)* found that plants with shorter flowering durations required smaller sample sizes to produce significant results when regressing flowering day on year.

Other studies such as *Calinger, Queenborough & Curtis (2013)* and *Lavoie & Lachance (2006)* disregarded the effect of flowering duration and noted the results of *Primack et al. (2004)*, which reported no bias associated with long or short flowering durations when mean estimations are analyzed. Plants in tropical regions often have long flowering durations (*Van Schaik, Terborgh & Wright, 1993*; *Fenner, 1998*), but as long as flowering is not continuous throughout the year, methods applied to temperate regions should also yield valuable insight into effects of climate change on phenology in the tropics. While studies using herbarium specimens to analyze long-term changes have been limited to temperate regions, future studies could use circular statistics to analyze long-term phenological changes in tropical regions (*Fisher, 1993*; *Morellato, Alberti & Hudson, 2010*). Circular statistics have been used to analyze flowering phenology in several tropical field studies, but these studies lacked long-term climate change analyses (*Novotny & Basset, 1998*; *Morellato et al., 2000*; *Cruz, Mello & Van Sluys, 2006*; *Rogerio & Araujo, 2010*; *Tesfaye et al., 2011*; *Nadia, Morellato & Machado, 2012*; *Nazareno & Dos Reis, 2012*; *Staggemeier, Diniz-Filho & Morellato, 2010*; *Carvalho & Sartori, 2015*; *Kebede & Isotalo, 2016*).

## Averaging temperatures

The foundational study by *Primack et al. (2004)* examined temperature averages from three calendar months prior to the specimen flowering date, with the assumption that flowering date is a function of temperatures experienced in past months. Field investigations such as *Fitter et al. (1995)* have shown temperature averages from different sets of months preceding flowering affected flowering phenology in different ways. More recently, *Calinger, Queenborough & Curtis (2013)* chose to regress the month of flowering with temperature averages from each of the eleven months prior to flowering. They found that temperature averages from three months prior to the date of flowering showed the strongest correlations with flowering (*Calinger, Queenborough & Curtis, 2013*). *Robbirt et al. (2011)* investigated three sets of temperature averages over three month intervals and also found that three months prior to flowering had the most predictive power. Similarly, *Rawal et al. (2015)* regressed flowering on temperature averages for each species from 1, 3, 6, 9, and 12 months prior to flowering, because responses can vary by species. *Rawal et al. (2015)* also found that mean temperatures three months prior had the greatest influence on flowering time for all species.

Other studies have used average temperatures from spring months because spring temperatures generally have the most predictive power for flowering date (*Miller-Rushing & Primack, 2008*; *Primack & Miller-Rushing, 2009*; *Robbirt et al., 2011*; *Calinger, Queenborough & Curtis, 2013*; *Park, 2014*; *Park & Schwartz, 2015*). *Bertin (2015)* found an interesting trend that supported the effect of spring temperatures: the earlier a species' mean flowering time occurred in the spring, the more the species' mean dates had shifted toward an earlier day of year over time. *Robbirt et al. (2011)* also found the highest correlations of flowering day with spring temperature averages across March, April and May. *Calinger, Queenborough & Curtis (2013)* found significant changes in flowering in response to average spring temperatures (February–May) but not in response to summer temperatures (June–September). *Gaira, Dhar & Belwal (2011)* found the highest correlations between flowering and temperatures in earlier months from December–February in a Himalayan perennial. As an alternative to using mean monthly temperatures, *Diskin et al. (2012)* investigated the averages of temperature anomalies, or deviations from the overall long-term mean, for 2, 3, and 6 month periods from January to June and found averages from six months prior to flowering had the strongest correlations. *Park (2014)* used temperature averages across three month periods from early spring to late summer and found a similar trend. Temperature averages were organized into early, mid, and late seasonal classes within the months of February–October. *Park (2014)* found warming temperatures had affected species in the early spring class more than other classes. *Park & Schwartz (2015)* also used early, mid and late seasonal classes for spring and summer and found that mid-season phenology events should be modeled differently than early or late season events. *Hart et al. (2014)* used annual temperatures and temperatures from each season (spring, summer, fall, and winter) and found significant correlations for annual and fall temperature averages, but with opposite effects. *Hart et al. (2014)* discussed that warmer fall temperatures may delay the chilling requirement for *Rhododendron* species, resulting in a delay in flowering while warmer annual temperatures will lead to advances in flowering overall. Other studies found annual temperature means were as useful as spring temperatures. *Davis et al. (2015)* found similar results between spring and annual temperature averages and used annual averages in analyses. *Gallagher, Hughes & Leishman (2009)* also used annual temperature means for analyses and explained that seasonal means were correlated with annual means.

We recommend investigating the effect of temperature by analyzing averages from multiple sets of months prior to flowering for each species rather than using only one fixed spring interval or only annual temperatures (*Diskin et al., 2012*; *Robbirt et al., 2011*; *Calinger, Queenborough & Curtis, 2013*). Caution should be taken when analyzing temperature averages from the same months prior to flowering for all species when flowering month varies by species. For example, when analyzing the effect of temperature averages from three months prior for all species, *Calinger, Queenborough & Curtis (2013)* found that for many species, flowering was correlated with temperatures three months earlier, yet for species with an earlier mean flowering day in April, January temperatures (three months prior to flowering) did not predict flowering date; instead, temperature averages from the months of February, March and April were better predictors for those species.

## Geographic variation

Among species that have broad geographic ranges, differences in climate in different parts of the species' range can complicate attempts to correlate a species' flowering day with temperature. Several methods have been used to account for climate variability across a species' range. An early study by *Lavoie & Lachance (2006)* investigated the effects of climate variation on the phenology of Coltsfoot (*Tussilago farfara* L.) across a range of about 10,000 km$^2$ in Quebec, Canada. Temperature data from 88 meteorological stations were averaged together across this range. To account for early snow cover melt in the southern part of this range, flowering dates from individuals in southern locations were normalized with individuals in northern locations by subtracting extra periods of snow cover from individuals in the north. The adjusted dates indicated flowering occurred 33 days earlier over the last century while original (unadjusted) dates indicated flowering occurred 19 days earlier over the last century.

While the study by *Lavoie & Lachance (2006)* adjusted actual dates for analyses, more recent studies mostly account for climate variation using georeferenced climate data at various scales. *Calinger, Queenborough & Curtis (2013)* accounted for climate variation across Ohio by using temperature averages from ten US Climate Divisions across the state, each about 8,000 km$^2$. A total of 344 Climatic Divisions were established across the contiguous United States in 1895 in order to monitor climate records more accurately. These divisions have now accumulated about 100 years of climate records (*Guttman & Quayle, 1995*). A later study by *Park (2014)* used average temperatures across the US county where each specimen was collected.

Other studies accounted for climate variation across longitude, latitude, or elevation. *Robbirt et al. (2011)* analyzed the geographical effect of longitude and found that flowering occurred 4.86 days earlier per degree of longitude in a westward direction across the southern coastal counties of England. A later study by *Bertin (2015)* used Hopkins' bioclimatic law to normalize dates on specimens. *Hopkins (1918)* generally stated that for every increase in a degree of latitude, or increase of 121.92 m elevation, the life history events of plants and animals were delayed by four days. *Bertin (2015)* found consistencies with Hopkins' bioclimatic law using latitude and elevation and chose to normalize flowering dates by adding expected phenological deviations from both latitude and elevation. *Gaira, Dhar & Belwal (2011)* also analyzed climate variation using elevation when temperature data were not available, assuming a 6.5 °C change in temperature per 1,000 m change in elevation in the Himalayan region.

Other studies used temperature averages across large regions. *Li et al. (2013)* used temperature data that was averaged from 36 meteorological stations across the Tibet Autonomous Region. *Molnár et al. (2012)* used temperature averages from 10 meteorological stations across Hungary and stated that the data were statistically indistinguishable across stations ($\sim$93,030 km$^2$). *Park & Schwartz (2015)* averaged temperatures from 13 stations across South Carolina, USA ($\sim$82, 931 km$^2$). A later study by *Robbirt et al. (2014)* used temperature averages from an area between Bristol, Preston, and London, across the United Kingdom ($\sim$17,000 km$^2$). *Robbirt et al. (2014)* used geographical divisions called Watsonian vice-counties specifically delineated for the

purposes of collecting scientific data, much like the US Climate Divisions. *Robbirt et al. (2014)* found temperature averages were sufficient because climate variation across the Watsonian vice-counties used in their study did not significantly differ.

In order to more accurately estimate temperature averages across a region, recent studies have used Geographical Information Systems (GIS) to project finer-scale climate layers across a region and extract temperature data from specific Global Positioning System (GPS) points. *Gallagher, Hughes & Leishman (2009)* referenced GPS locations for each specimen and extracted the temperature averages at specimen GPS points from a gridded map of temperature averages across Australia ($\sim$5 km$^2$ resolution). *Hereford, Schmitt & Ackerly (2017)* also extracted climate data from 176 collection locations in order to analyze species distributions and phenology. *Rawal et al. (2015)* used the nearest data point from gridded climate averages across Victoria, Australia. *Edwards & Still (2008)* analyzed the climate envelopes of grasses by assigning GPS points to herbarium specimen locations in order to extract temperature averages from gridded climate maps (250 m$^2$ resolution). *Kosanic et al. (2018)* manually geo-referenced locations using herbarium specimen localities and provided a methodology for assigning GPS coordinates when analyzing species distributions and phenology. Standardizing methods for geo-referencing localities of herbarium records without GPS coordinates could allow for more specimen data and larger sample sizes. *Bloom, Flower & DeChaine (2018)* developed a comprehensive protocol for standardizing spatial accuracy of geo-referenced specimen localities for species distributions.

Future studies of phenology could benefit from such geo-referencing methods because several phenology studies only included data from specimens with GPS coordinates. Studies using GPS data are able to account for climate variation with higher resolution, although accuracy still depends on the underlying empirical data and modeling approach used to generate GIS climate layers.

We recommend using the most spatially precise temperature data available, such as climate divisions (*Calinger, Queenborough & Curtis, 2013*; *Robbirt et al., 2014*) rather than state or region averages (*Li et al., 2013*; *Park & Schwartz, 2015*). Using GPS specimen data to identify local climate conditions from GIS climate layers (*Gallagher, Hughes & Leishman, 2009*; *Edwards & Still, 2008*) is also now generally more precise and convenient in comparison to making generic and coarse-scale corrections for latitude, longitude or elevation (*Gaira, Dhar & Belwal, 2011*; *Robbirt et al., 2011*; *Bertin, 2015*). If temperature averages from larger areas are used, we recommend testing for climate variability across smaller divisions before using averages across the larger area (*Lavoie & Lachance, 2006*; *Molnár et al., 2012*; *Robbirt et al., 2014*).

## Validation: herbarium specimens versus field observations

Field data are often combined with herbarium specimen data in analyses, allowing for comparison and sometimes allowing for validation of conclusions based on herbarium data (*Primack et al., 2004*; *Miller-Rushing et al., 2006*; *Bertin, 2015*). *Primack et al. (2004)* used herbarium specimens for historic data and field observations for current data and combined the two in analyses. Studies by *Miller-Rushing et al. (2006)* and *Bertin (2015)* also compared herbarium specimen data with field observations. *Miller-Rushing et al.*

*(2006)* found that phenology inferences from herbarium specimens alone differed from the combined data by only about one day.

An early study by *Borchert (1996)* found that herbarium specimen data produced slightly longer flowering durations than field data, but noted that durations were mostly similar overall. *Borchert (1996)* and *Rivera & Borchert (2001)* found phenology data from field sites largely overlapped that of herbarium specimens with only slight differences. The negligible differences between herbarium specimen data and field data in these studies helped justify the use of herbarium specimen data to analyze phenology in more recent studies. Nevertheless, several more recent studies specifically compared phenology estimates from field data to those made from herbarium specimens.

*Bolmgren & Lonnberg (2005)* compared herbarium specimen data directly to field data and found the two data sets were overall highly correlated with only minor differences. For example, herbarium specimens showed a slightly earlier mean flowering for spring-flowering plants than field data, but the difference was not significant (*Bolmgren & Lonnberg, 2005*). Later studies by *Robbirt et al. (2011)* and *Davis et al. (2015)* also primarily focused on testing the validity of using herbarium specimen data. *Robbirt et al. (2011)* used a principal axis regression analysis to compare herbarium derived peak-flowering dates with field derived peak-flowering dates and found a high degree of correlation. *Robbirt et al. (2011)* discussed how the high degree of correlation between herbarium and field data also supports the notion that geographically different records will not significantly alter the robustness of either data set. A study by *Davis et al. (2015)* used a paired $t$-test to compare mean first-flowering day between herbarium specimens and field data and found no statistical difference. *Davis et al. (2015)* concluded that both specimen and field data could be combined and used as a whole.

In order to increase sample sizes, *Molnár et al. (2012)* added about 2,000 field observations to about 5,000 herbarium records, resulting in 70% herbarium records for the study. Similarly, *Panchen et al. (2012)* added about 2,000 field records to about 1,500 herbarium records, for a total of 43% herbarium records for the study. *Searcy (2012)* combined herbarium specimen and field data and then split the combined data into two time periods (1863–1935 and 1994–2008). Herbarium specimen data may provide some advantages over field data. *Bolmgren & Lonnberg (2005)* and *Primack et al. (2004)* noted that using herbarium specimens conserves time and resources, especially when species are located in difficult to access geographical areas such as mountain peaks or islands. Herbarium specimens are also collected over a greater period of time from a larger geographical area while field data are often from specific localities over a shorter time period (*Primack et al., 2004*; *Bolmgren & Lonnberg, 2005*; *Bertin, 2015*; *Davis et al., 2015*). Herbarium specimens also provide long-term records that are widely accessible for multiple studies. Despite criticisms, herbarium specimen data have been shown to produce similar enough results to field data that herbarium specimen data are now widely accepted in phenological studies.

## CONCLUSIONS

The use of herbarium specimens for the investigation of flowering phenology has grown considerably during the past decade. As efforts to produce digital copies of specimens and label information have amassed large datasets, new approaches for analyzing responses to climate change are rapidly becoming available. Although small sample sizes have often been used in early studies of phenology, various factors, such as wide geographic range, may require larger sample sizes. Based on recent validations, estimations of mean-flowering should be used rather than first-flowering because estimates of first-flowering are more sensitive to sampling. Statistically modeling the start of the flowering process appears to be another promising approach to investigating how climate change has affected the beginning of a flowering cycle (*Pearse et al., 2017*). The use of GPS data appears to be the way forward for the advancement of methods in the study of phenology. GPS point data allow for correspondence with higher resolution temperature data in climatically diverse geographical regions. Studies using herbarium specimen data will continue to help us understand the impact of recent climate change on plant reproductive phenology. Other aspects of plant phenology that can be analyzed using herbarium specimens, such as fruit ripening and spring leaf emergence, have important implications for higher trophic levels, which may include rare animals dependent on plant resources (*Everill et al., 2014*; *Zohner & Renner, 2014*; *Mendoza, Peres & Morellato, 2016*). Studies using herbarium specimens have become an asset for long-term climate change vulnerability assessment. These studies have begun to analyze the effects of climate change on community composition (*Miller-Rushing & Primack, 2008*; *Park, 2014*), species distribution (*Hereford, Schmitt & Ackerly, 2017*; *Kosanic et al., 2018*), coevolved plant pollinator relationships (*Molnár et al., 2012*; *Robbirt et al., 2014*), functional groups (*Miller-Rushing & Primack, 2008*; *Panchen et al., 2012*; *Calinger, Queenborough & Curtis, 2013*; *Munson & Long, 2017*), and phylogenetic relationships (*Bolmgren & Lonnberg, 2005*; *Molnár et al., 2012*; *Primack & Miller-rushing, 2012*). Future studies investigating phylogenetic signals and plasticity are needed to further improve our understanding of adaptation and resilience to climate change. As temperatures continue to rise globally, herbarium specimens will continue to be crucial resources for analyzing phenological responses to climate change.

### Funding

The authors received no funding for this work.

### Competing Interests

Curtis C. Daehler is an Academic Editor for PeerJ.

### Author Contributions

- Casey A. Jones conceived and designed the experiments, performed the experiments, analyzed the data, contributed reagents/materials/analysis tools, prepared figures and/or tables, authored or reviewed drafts of the paper, approved the final draft.

- Curtis C. Daehler prepared figures and/or tables, authored or reviewed drafts of the paper, approved the final draft.

## Data Availability

The research in this article did not generate any data or code (literature review).

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
