# Peer review of "Herbarium specimens can reveal impacts of climate change on plant phenology; a review of methods and applications"

_PeerJ, doi:10.7717/peerj.4576_

## Round 0.1 · original submission · Major Revisions

Reviewer number 2 raised a number of issues on the representativity of the study and pointed out that there was a number of relevant literature missing.

I would like to point out that the number of studies that entered the "systematic review" is quite modest. Therefore, I would argue that the advantages of "subjective potentially biased selection" do outweigh the advantages of an "objective systematic selection".

Also, I would state that the study does not have many elements of a "meta-analysis" i.e. does not do inference about the subject matters based on a statistical analysis. I would therefore remove all references to the word systematic review for your article.

·

Basic reporting

English writing needs improvement to make the writing concise. Please remove redundant information. More detailed comments were provided after general comments.

Experimental design

No comment

Validity of the findings

No comment

Additional comments

This study summarized the approaches and applications of using herbarium specimens to analyze the changes in flowering phenology in the last decades. This is valuable work. The general and detailed comments are provided as below.

General comments:
(1) The authors need to briefly introduce how the phenology data (such as mean flowering day of year or first flowering day of year) are obtained from the herbarium specimens. Are they the dates when herbarium specimens were collected? This is important information for readers who are unfamiliar with this type of data.

(2) Many studies have shown that the inferences from herbarium specimen data are comparable to findings from field observations. What are the advantages and disadvantages of using herbarium specimens than field observations, especially in the future when field observations are being recorded more? Do herbarium specimens provide any extra data than field observations? Please briefly discuss it.

Detailed comments:
L55: “temperature regressed on year” does not represent the changes in phenology. Please rephrase this sentence.
L56: “but other approaches may be suitable in some cases” Please clearly point out what are “other approaches”, and “some cases”.
L64-66: the last sentence in the abstract is loosely related to the approach reviewed by this study. Please revise it to focus on the important role of herbarium specimens in studying the changing rate of phenology under continued warming.

L69: please provide the reference for Carl Linnaeus’s study of phenology.
L71: please provide the original reference for “Charles Morren who introduced the term around 1850”
L84: 2017 -> 2015?
L82-84: Please explain why herbarium specimens may be useful for these different cases.
L94: “model set”? Do you mean “procedure”?
L97: correlations of what? Change of what? Please specify.
L104: “or some combination of these with multiple regression models”. Please explain or delete.
L105. This could be more concise. For example, “whether flowering day was earlier in warmer years, whether flowering day got earlier over years.”
L119: Is there any reason herbarium specimens were not used in tropical region? I don’t think the availability of herbarium specimens data is one reason.
L122: provide -> provided
L130: This is redundant: “we only selected studies that met the all three eligibility criteria”. Because you just referred “Studied were eligible for this review if they met three main criteria”
L164: some significant trends? Vague expression. Please specify.
L354: discusses -> discussed

Reviewer 2 ·

Basic reporting

.

Experimental design

.

Validity of the findings

.

Additional comments

This is a well-written and well-organized review of the plant phenology. This paper will serve as a useful introduction to the field for researchers who want to start conducting research in the field, and effectively lays out the basic types of analysis in this field. Over the past few months, there have been several important papers in this field that make this paper seem somewhat dated. First, Willis had an article in TREE that outlined how herbarium specimens could be used in climate change, now that millions of specimens are available on-line. Second, Pearse had an article in Nature Ecology and Evolution describing a new statistical technique for combining peak flowering dates from herbarium specimens with first flowering observations in the field. And third, Daru had an article in New Phytologist about the biases of herbarium specimens. This article needs to be revised in light of these three recent articles.
The article describes the use of monthly average temperature data to compare to with flowering times. However, many researchers in the field prefer to use growing degree data. That said, a review article by Basler demonstrated that both types of analysis yield almost identical results.
In the article, the authors suggest that mean flowering date is better than first flowering date because first flowering date is more variable. However, the reason that first flowering dates are problematic is that they are affected by many sampling issues, especially changing population size over time. Populations that decline over time will have later first flowering dates even with the mean is the same.
A key concluding point is that the herbarium specimens could be used for other phenological characters, such as leaf out times. The authors seem to be unaware of the two 2014 papers that do exactly that by Zohner and Renner in Ecology Letters and by Everill and others Am J Bot.
Finally, a key point made by the Willis paper which needs to be emphasized is that this field is ready to undergo a huge expansion now that millions of herbarium specimens are becoming available on-line.
If the authors can revise their paper with these points in mind, this paper could become a useful contribution.

---

## Round 0.2 · Minor Revisions

I have been rereading the article and I have been trying to recontact the more critical reviewer (reviewer 2). Since I have not heard from them I decided to take the "law into my own hands".

I was generally satisfied with your corrections. However, two points come into my mind. Firstly, the article has been changed from a systematic review to a review of the literature. This makes (a) the selection criteria less stringent and it would be nice to add a few additional papers from the literature which you did not find with your initial search words (b) the prism diagramme etc are linked to meta-analysis and systematic reviews but not to "more free and less stringent literature analysis". I would suggest to remove the prism diagramme. My reason is that the prism-diagramme is linked to publication biases and selection biases in quantitative analysis. However, your paper is a qualitative analysis and these diagrammes are therefore meaningless.

---

## Round 0.3 · accepted · Accept

I enjoyed reading your article